# A Review of Selective Laser Trabeculoplasty: “The Hype Is Real”

**DOI:** 10.3390/jcm11133879

**Published:** 2022-07-04

**Authors:** Tomislav Sarenac, Anela Bečić Turkanović, Peter Ferme, Tomaž Gračner

**Affiliations:** 1Department of Ophthalmology, University Medical Center Maribor, Ljubljanska 5, 2000 Maribor, Slovenia; dr.tomislav.sarenac@gmail.com (T.S.); anela.becic@gmail.com (A.B.T.); pferme@gmail.com (P.F.); 2Faculty of Medicine, University of Maribor, Taborska Ulica 8, 2000 Maribor, Slovenia

**Keywords:** glaucoma, laser treatment, trabecular meshwork, dropless treatment, intraocular pressure

## Abstract

Presently, there is no efficacious treatment for glaucomatous optic neuropathy; the current treatment is focused on lowering intraocular pressure (IOP). Studies have demonstrated the safety and efficacy of selective laser trabeculoplasty (SLT) in reducing the IOP in eyes with open-angle (OAG) glaucoma or ocular hypertension (OH). Moreover, the European Glaucoma Society has instated SLT as the first-line or adjunctive treatment in OAG or OH, reiterating its clinical significance. In this review, we outline the old and the new roles of SLT, with an emphasis on clinical practice, and look further into its renewed appeal and future developments.

## 1. Introduction

Glaucoma is the third largest cause of blindness worldwide, after unaddressed refractive errors and cataracts [1]. The global prevalence in the elderly population worldwide is estimated at 3.5%. It was presumed that by 2020 there would have been 79.6 million people affected with glaucoma; this number might increase to 111.8 million globally by 2040, causing a significant decrease in the quality of life and economic burdens [2]. It is assessed that, currently, 57.5 million people are affected by primary open-angle glaucoma (POAG) [3]. The main goals of glaucoma treatment are to preserve visual functioning (adequate to individual needs), with minimal or no side effects, for the expected lifetime of the patient, without any disruption of normal activities, at a sustainable cost [4]. Glaucoma is a disease associated with optic nerve degradation (glaucomatous optic neuropathy), which causes visual field loss, and is responsible for significant visual morbidity, i.e., loss of independence. Presently, there is no proven efficacious treatment of glaucomatous optic neuropathy. Therefore, the treatment is focused on reducing intraocular pressure (IOP), which is the only risk factor linked to glaucoma progression that can be successfully influenced [5]. Reducing IOP can be achieved with medical, surgical, or laser treatments. The most common initial treatment is with hypotensive drops; however, patient adherence to a treatment regimen could be relatively low [6]. In 1998, the first successful protocol of selective laser trabeculoplasty (SLT) was established, a 532-nm Q-switched frequency-doubled Nd:YAG laser with a single pulse of short duration and low fluence was used and has become an established method for lowering the IOP in the treatment of open-angle glaucoma (OAG) and ocular hypertension (OH) [7]. It targets the trabecular meshwork (TM), which improves aqueous outflow, contributes to reducing IOP, and does not require extensive patient compliance.

Multiple studies have, to a high extent, demonstrated the safety and efficacy of SLT in reducing IOP in OAG or OH. However, most of the studies have reported on SLT as an adjunctive treatment [8,9,10,11]. This has left the role of primary SLT somewhat ambiguous; however, it appears to be vastly more important in clinical practice than ascribed in the guidelines. SLT could be considered one of the cornerstones of dropless glaucoma therapy in newly diagnosed OAG or OH [4]. This has recently been further upheld by randomized controlled trials, supporting the case for SLT as the first-line treatment of glaucoma, such as in the laser in glaucoma and ocular hypertension (LiGHT) study [12]. The European Glaucoma Society Terminology and Guidelines for Glaucoma, 5th Edition, recently listed SLT earlier in the algorithm of glaucoma treatment [13]. It was stated that SLT can be used sooner, as an alternative to failed first monotherapy, as a single glaucoma treatment, or as an adjunctive treatment later on; this has renewed the appeal of SLT to clinicians.

In the following review, we will outline some of the crucial clinical guidelines for SLT, especially in OAG and OH, and concisely provide useful data to provide more information about this topic, focusing on recent relevant studies. Our search for studies was conducted using the PubMed database; the search strategy is available in Appendix A. Finally, promising future directions in this area will be introduced, with an outline of novel clinical studies.

## 2. Basic Principles

Although lasers have gained great popularity in glaucoma management over the last two decades, the history of laser treatment for glaucoma started back in the early 1970s with Q-switched laser goniopuncture being the first technique described [14,15]. While the technique succeeded in IOP reduction, success was short-term. A few years later, argon laser trabeculoplasty (ALT) was presented by Wise and Witter [16]. They postulated a mechanical mechanism, in which laser-induced thermal burns of the TM caused collagen shrinkage following scarring of the TM. This tightens the corresponding meshwork and reopens the adjacent, untreated intertrabecular spaces, facilitating aqueous outflow. Ultrastructural TM modifications occurred before the IOP-reducing response, suggesting the mechanism of action is more complex. The cellular theory proposes that, in response to coagulative necrosis induced by the laser, there is elevated cytokine production, causing remodeling of the juxtacanalicular extracellular matrix, a likely site for the aqueous outflow resistance, improving the outflow facility [17,18].

ALT causes IOP reduction through increased aqueous outflow, confirmed by both tonography and aqueous dynamic studies [19]. With 30% IOP reduction, ALT was presented as a first-line therapy and as a second-line therapy [20]. Adverse events related to ALT were transient acute IOP spikes following the laser, development of peripheral anterior synechiae (PAS), corneal endothelial changes, and acute anterior uveitis [21]. Although serious side effects rarely occurred, most of the authors reported falling effects over time [22,23]. Latina and Park first introduced selective laser trabeculoplasty in their in vitro study in 1995 [24]. Using Q-switched frequency-doubled 532 nm Nd:YAG laser SLT targets the pigmented TM cells selectively without damaging the adjacent non-pigmented cells or other structures of the TM [25].

## 3. Mechanism of Action

The mechanism by which SLT lowers IOP is not completely understood and is likely multifactorial. SLT is based on the principle of selective photothermolysis first described by Anderson and Parrish 1983, in which radiation energy applied to the TM selectively targets pigmented cells without causing thermal damage to adjunctive structures [26]. Latina and Park demonstrated the SLT effect by selectively targeting pigmented TM in their in vitro [24] study on bovine TM cell cultures, and a few years later, in their in vivo study [7]. The extent of pigmented cell depletion after SLT depends on the magnitude of the energy used and the distance from the center of the irradiated zone reported by Wood et al. in their in vitro study [27]. In 2001, Kramer and Noecker reported less structural damage to the human TM in SLT-treated eyes compared to ALT in their in vitro study [28]. In 2003, Cvenkel et al. compared histopathological changes occurring in the eyes after ALT and SLT in their in vivo study and reported a smaller extent of the damage to the TM after SLT [29]. A meta-analysis comparing ALT to SLT revealed similar efficacy in the therapeutic IOP response. However, SLT has resulted in a greater reduction in the number of glaucoma medications versus ALT [30,31]. Moreover, SLT appears to be more effective in IOP reduction in retreatment versus ALT [32]. The authors report SLT’s effect in lowering IOP by increased outflow through TM [33,34] without significant differences in the aqueous humor dynamics comparing Caucasian and African races [35]. Vikas et al. discovered the IOP-lowering effect of SLT being mediated through an increase in the outflow facility using fluorophotometry and tonography in their study. They suggested higher aqueous flow and a lower outflow facility as predictive factors for better response to SLT [36]. As described, structural damage occurring to the TM in ALT is not detected in SLT patients; therefore, the mechanical and structural theories that have been suggested to explain ALT’s mechanism of action do not fully apply to SLT [37]. Furthermore, the biological theory of SLT action proposes that the laser modifies cellular activity by cytokine release, facilitating aqueous outflow [38]. Lee et al. revealed that the matrix metalloproteinase release was pigment-dependent and was not detected in non-pigmented cells after SLT [39]. The biological and biochemical changes have been observed in the TM after SLT. Alvardo et al.’s in vitro study reported a substantial increase in the number of monocytes/macrophages in the TM after SLT, resulting in the outflow facility augmentation and conductivity of human Schlemm’s canal endothelial cells [40].

Bradley et al. used the human anterior segment organ cultures, subjected them to laser trabeculoplasty, and detected increased stromelysin expression provoked by elevated IL-1 beta and TNF-alpha, which work synergistically [41], resulting in remodeling of the juxtacanalicular extracellular matrix and restoring normal outflow facility [17].

Izzotti et al. published a study aimed at the gene expression changes induced in TM cells by SLT using hybridization on miRNA-microarray and laser scanner analysis [42]. The study showed expression modulation of genes involved in cell motility, intercellular connections, extracellular matrix production, protein repair, DNA repair, membrane repair, reactive oxygen species production, glutamate toxicity, antioxidant activities, and inflammation. Regulation of aqueous humor outflow from the anterior chamber was reported to be modulated with SLT at the postgenomic molecular level without inducing damage at molecular or phenotypic levels.

## 4. Indications and Preoperative Evaluation

From a pragmatic clinical standpoint, we divide therapeutic indications for SLT into three groups.

The first group involves patients with POAG or OH without any prior glaucoma treatment, where SLT can be used as a primary (first-line) therapy. Most studies have compared SLT efficacy against topical medication and have found similar IOP-lowering efficacy. The LiGHT trial showed that 74.6% of eyes treated with primary SLT achieved drop-free disease control at the 3-year follow-up and has a comparable IOP reduction and complication profile to MIGS with smaller anatomical changes to the angle, and can therefore be recommended as an alternative or a first step treatment [43,44].

The second group involves patients with POAG or OH (with uncontrollable IOP and disease progression) who are already receiving glaucoma treatment, where SLT can be used as adjunctive therapy. Studies have shown that SLT successfully lowers IOP in (the eyes of) patients who are on hypotensive medication, have undergone previous ALT treatment, or have had glaucoma surgery [4,7,45,46]. SLT can also be repeated with an IOP reduction similar to the first treatment, or be used to delay glaucoma surgery [47,48,49].

The third group is patients with POAG or OH on glaucoma medication with adequate IOP control and without glaucoma progression, where SLT can be used as replacement therapy, i.e., to lessen the burden of medications. Since drops require strict daily dosing and have many side effects, adherence to medication is often poor. Treatment with SLT in patients already treated with glaucoma medication can lead to better IOP. A study by Lee et al. showed that patients treated with SLT require fewer medications to maintain their IOP goals [50]. In a study by De Keyser et al., SLT was able to completely replace medical therapy in 77% of patients’ eyes after 18 months, and may severely reduce local and systemic side-effects commonly caused by medication [51].

However, in the published literature, indications for treatment are most commonly divided by glaucoma type. The majority of studies have focused on SLT treatment in POAG and OH, but it is increasingly being used in other glaucoma types. When used in patients with pseudoexfoliative glaucoma, SLT shows similar IOP reduction to POAG [38,52,53,54,55]. In pigmentary glaucoma, the results of using SLT are similar, but there seems to be an increased rate of postoperative complications, probably due to a higher TM pigmentation and greater energy absorption [56]. Normal-tension glaucoma has lower baseline IOP so the IOP reduction is proportionally smaller [50,57]. SLT has also been used in primary angle-closure glaucoma where it has shown comparable IOP reduction to POAG, but at least 180° of the TM has to be visible and patients have to have an open laser iridotomy [58,59]. SLT has also shown promising results in treating steroid-induced glaucoma [60,61].

SLT is contraindicated when the TM cannot be visualized (e.g., angle closure, anterior synechiae, corneal opacity, poor patient cooperation, etc.). Even though there is a study that suggests that it is safe to perform SLT in patients with uveitic glaucoma, it should absolutely be avoided in active uveitis and be reserved only for the most refractory cases [62]. According to mechanisms of action, SLT is not suited for neovascular and congenital glaucoma treatment, where IOP cannot be decreased by TM outflow modification, although successful cases have been reported in pediatric cases of POAG of different pathophysiologies, with normal angles [63].

## 5. Operative Technique

To assess if the patient is an eligible candidate for SLT, a thorough glaucoma evaluation has to be made before treatment. Special importance has to be given to gonioscopy, where the visibility and pigmentation of the TM have to be evaluated.

Studies have mostly shown that perioperative topical medications lower the risk of an IOP spike but there is no consensus on what the best prophylactic treatment is [64]. Most practitioners recommend using a topical alpha-adrenergic agonist (apraclonidine or brimonidine) 15 min to 60 min before treatment; some practitioners also use miotic drops (1% to 4% Pilocarpine). A topical anesthetic is given and a gonioscopic contact lens is selected, preferably one without a laser spot magnification. There are plenty of lenses made especially for SLT. A coupling gel should be used. A 400-micron spot size and 3 ns pulse duration are standard for SLT. The aiming beam is pointed over the entire width of the TM. The initial power is normally set at 0.8 mJ, but should be lower in heavily pigmented meshworks (e.g., 0.4 mJ), since side effects can be more severe if a higher power is used [56]. The power is then increased or decreased until the minimal power to form small cavitation bubbles is acquired (threshold power), and then decreased by 0.1 mJ to set the power used for treatment. Some practitioners prefer to treat with the threshold power though; 25–100 adjacent (but not overlapping shots) are applied over 90°–360° of the meshwork, depending on the protocol used. It is advisable to always treat the same quadrants or halves first (for example, the bottom half), so that if retreatment is performed, it can be conducted on the other (previously untreated) half. Immediately after treatment, another drop of alpha-adrenergic agonist can be given. IOP should be remeasured 30 to 60 min after treatment; if it is elevated, additional medication may be needed and a closer follow-up planned.

Many studies and meta-analyses have compared the treatment of different degrees of the TM. While some found a significant difference in the IOP-lowering effect between treating 180° and 360° of the TM in POAG [34,65], others did not [66,67,68], but one study showed lesser diurnal IOP fluctuations when treating 360° [69]. Most reviews have concluded that there is no significant difference when treating 180° or 360° [11,38,70], as confirmed by a recent meta-analysis [71]. Studies have also researched different power levels, mostly finding that a higher power leads to greater IOP reduction (but also more adverse effects) [72,73].

## 6. Postoperative Management

There is an ongoing debate regarding the best peri- and post-operative treatments, and many studies have attempted to establish the best practices, mostly with contradicting results. The main adverse effects are post-operative IOP elevation (IOP spike) and anterior chamber inflammation. Depending on the practitioner and the patient (e.g., baseline IOP, advanced glaucomatous damage), topical anti-inflammatory and IOP-lowering drops are commonly prescribed for 4–7 days but are often not needed.

IOP spikes can occur after SLT, especially in high-risk patients and they typically arise within 24 h. Zhang et al. [64] analyzed 22 randomized clinical trials (1 SLT and 21 ALT trials) and concluded that the use of perioperative IOP-lowering medications is superior to no medications in preventing IOP spikes after laser trabeculoplasty, with little to no adverse effects. Apraclonidine, brimonidine, acetazolamide, and pilocarpine are commonly used. If SLT is used as an adjunctive treatment, existing glaucoma medication is typically continued.

Another important factor is managing postoperative inflammation. Because of a typical anterior chamber inflammation seen after ALT, most practitioners routinely prescribe anti-inflammatories, especially steroids, and the practice continues with SLT. Treatment with steroids or NSAIDs has not shown a significant decrease in postoperative anterior chamber inflammation [74]. Since ALT and SLT have different mechanisms of action, questions about the long-term effects of post-SLT inflammation on the IOP-lowering effects remain.

One of the mechanisms of action in SLT is thought to be the activation of inflammatory pathways that cause TM remodeling and better functioning of the TM, with increased outflow and a reduction of IOP [33,36]; therefore, a possible contra-productiveness of using anti-inflammatory medication was proposed. On the other hand, inflammation may cause fibrosis and scarring, restricting outflow and thereby decreasing SLT efficacy, a mechanism that anti-inflammatory medications could partially prevent. Most of the studies found no benefits in postoperative treatments with anti-inflammatory drops, especially in patients with lower baseline IOP [74,75,76,77]. Surprisingly, the steroid after laser trabeculoplasty (SALT) [78] study found significantly better IOP reduction at 12 weeks in patients’ eyes treated with steroid or NSAID drops after SLT (compared to the placebo) and, therefore, contradicts most previous studies.

It is therefore not possible to establish a clear protocol for postoperative management. After reviewing the available literature, our conclusion is that it should be individually tailored to the patient (baseline IOP, glaucoma risk, previous medication, or surgery) and the performed treatment (degree of trabecular meshwork treatment, energy used, etc.).

## 7. Outcomes

There are numerous studies that contribute to the topic of SLT and its outcomes. SLT has mainly been compared to monotherapy or is used as an adjunctive treatment in various types of glaucoma patients. Here, we concisely provide outlines of the clinically most relevant recent studies for SLT as a first-line therapy or as a means for lowering the dependence on drops or adjunctive therapy. In this section of our review, the participants were mostly patients with POAG and OH, albeit SLT could be effectively used in another OAG, such as pseudo-exfoliative or pigmentary [11].

The hype that SLT can challenge medical therapy as a first-line treatment was materialized with the LiGHT trial, which compared the cost-effectiveness, efficacy, and safety of SLT versus hypotensive medical therapy for the initial treatment of glaucoma. The authors concluded that ‘SLT should be offered as a first-line treatment for open-angle glaucoma and ocular hypertension’ [12]. This randomized controlled trial (RCT) was one of the largest to date and was diligently designed around putting SLT first in a real-life glaucoma practice. Another reason why this trial stands out is because of the definition of target IOP reduction from baseline. Unlike the majority of studies, where >20% reduction in IOP was targeted, a more customizable approach was taken. Target IOP was established according to each patient’s glaucoma severity, and the target was modifiable during the study. In cases of glaucoma progression, despite IOP being targeted, the target IOP was further lowered, and vice versa in cases where no progression was detected. The follow-up and adjunctive treatment were similarly determined according to glaucoma progression. In our opinion, this trial setting contributes to the real-world character and subsequently provides more clinically relevant data. On the other hand, this could be considered less stringent, since “reaching the target” did not necessarily coincide with >20% reduction in IOP as strived for in most other studies reviewed. This might have contributed to the high success rate of the SLT-first group. In the trial, treatment-naïve POAG and OH patients were stratified into the medication-first group and the SLT-first group. In the SLT group, 95% of patients reached the target IOP at 36 months of this 78.2%, with no adjunctive medication, whereas in the medication group, 93.1% reached the target, with 64.6% requiring only prostaglandins, which were prescribed as a first choice. The difference was perhaps most striking in the number of trabeculectomies, where none of the 356 patients in the SLT-first group needed surgery and 11 of 362 patients in the medication-first group needed incisional glaucoma surgery. Furthermore, during the study period, less treatment escalations were observed in the SLT-first group. Transient side effects, such as discomfort and hyperemia, were common (34%); however, they were temporary in contrast to the known side effects of hypotensive medication. This trial contributed significantly to considering SLT as a first-line glaucoma treatment, with nearly no side effects, which translates to treatment efficiency, especially regarding patient compliance with medical therapy. A further post hoc analysis has shown similar results for POAG and OH patients [79], where approximately 75% of patients reached dropless IOP control at 36 months after primary or repetitive SLT, with the majority achieving the target after the first SLT. Regarding glaucoma progression (in terms of visual field testing), it has been shown that patients in the SLT- first group were less likely to have rapid visual field progression [43].

In a retrospective study by Ansari [80], with a 10-year follow-up success rate of 72% (at 10 years), with visual field loss remaining stable, 60% required retreatment in 10 years. Here, the success rate as the main outcome measure was defined as an >20% reduction in IOP and IOP < 19 mmHg. In addition, no patients in the study needed trabeculectomy at 10 years akin to results from the LiGHT trial. Albeit, the LiGHT trial has not shown significant improvements in health-related quality of life compared to medical therapy, we agree with Ansari, who stated that longer-term data from their study could imply substantial improvements in quality of life, most likely regarding medication avoidance, possible toxic effects, and costs. This matter was also studied by Ang et al. [81], where the quality of life was no different between naïve-treated SLT or topical medication; however, it was reported that a higher proportion of patients with eyelid erythema and conjunctival injection were found in the medication-only group.

One recent meta-analysis by Chi et al. [71] on SLT treatment in naïve patients versus medication with 1229 patients has reported no difference in treatments with SLT and medication-only treatments regarding the IOP reduction. Furthermore, SLT was slightly more effective when the medication-only group was taken as a reference, with 180-degree SLT performing slightly better than the rest of the trabeculoplasty methods analyzed (albeit these differences were not significant). Furthermore, Chi et al. showed that patients who underwent SLT and needed drops ultimately required less medication than the medication-only group. These findings are in accordance with other metanalyses we found [31,81,82]. In the meta-analysis by Zhou et al., where different modalities of laser trabeculoplasty were studied on 2859 eyes, they found 180-degree SLT to be somewhat more effective at reducing the number of medications needed in comparison to ALT, whereas no difference was found between five other modalities (270-degree SLT, 360-degree SLT, new laser trabeculoplasty, transscleral 360-degree SLT without gonioscopy, and low-energy 360-degree SLT). All of the above have demonstrated equal effectiveness for IOP decreases in comparison to hypotensive medications [31].

We believe that real clinical data, collected during every-day clinical practice, adds to the relevance of SLT, to some extent, when validating the results from trials and metanalyses. However, in the real world, separating the effects of SLT from the effects of coexistent hypotensive medication in patients is nearly impossible. Up until now, simultaneous use of therapies usually occurred in the average glaucoma practice. Two of such real-world data reports of retrospective studies on SLT have been published recently and have shown somewhat fewer persuasive results.

Khawaja et al. published a study that was conducted in the United Kingdom (UK); they demonstrated that 70% of eyes responded to SLT treatment at 6 months, but success by 2 years was sustained only in 27% of cases [83]. The Kaplan–Meier survival analysis has shown that 83% of eyes could fail at 36 months. The measures of failure could be considered stringent by some, e.g., an inadequate reduction in IOP (>21 mmHg or <20% reduction), an increase in the number of glaucoma medications, or a subsequent glaucoma procedure, including repeated SLT. Efficacy of SLT was higher in cases with higher baseline IOP (IOP > 21 mmHg) and was not altered by the severity of glaucoma or the coexistent use of hypotensive medication. In cases of higher baseline IOP, there was a 32% lower risk of failure compared to the (eyes of) patients with IOP ≤ 21 mmHg at baseline. It could be extrapolated that SLT is more effective in OH or high-IOP OAG than for normal-tension glaucoma. Mostly, patients were on prostaglandins and no association to SLT failure was found when compared to the rest of the hypotensive medication used. Patient selection was not as rigorous as in LiGHT and the metanalysis by Chi et al. In those publications, naïve mild glaucoma patients with no concurrent ocular diseases were included (visual field not worse than −12 dB in the better eye on the Humphrey field analyzer in the LiGHT trial).

The following study by Abe et al. [84] revolved around similar endpoints and reported significantly better results regarding SLT efficiency. SLT was studied for three common indications: uncontrolled IOP without medications, uncontrolled IOP with medications, and controlled IOP with medications for the purpose of reducing the number of hypotensive medications. Treatment failure was considered in the following cases: subsequent procedures (including SLT), IOP > 21 mmHg or IOP reduction < 20%, and an increase in the number of different glaucoma drops. A total of 54.7% failed according to these criteria during the 36-month follow-up. When the Kaplan–Meier survival analysis was stratified according to the indications listed above, SLT as a first-line treatment had 80% success at 12 months, which decreased to 46% at 36 months. The commonest scenario in the study was SLT in patients with medically well-controlled IOP, with an intention to lower the number of drops taken (55%). In this group, 49% had success with SLT at 36 months and 37% remained dropless for 36 months. This implies that SLT is a valid tool to reduce the number of hypotensive medications. Denser angle pigmentation, corticosteroid treatment following SLT, and earlier stage glaucoma were associated with lower risks of failure. The latter reiterates the concept that SLT is a valid option as a first-line treatment, especially in early, mild glaucoma compared to patients with advanced glaucoma.

## 8. Complications

SLT is considered a safe procedure and is well-tolerated by patients with low complication rates, ranging from 0% to 65.7% [81,85]. Complications associated with SLT are mostly transient and self-limiting, such as momentary mild redness, discomfort or mild pain, anterior chamber inflammation, or an IOP spike in the first week. The LiGHT trial reported SLT as a safe method, preserving its safety frame in the procedure’s repetition [47]. Although this study reported only self-limiting adverse effects of lasers, there are some uncommon and severe complications, such as transient corneal thinning, endothelial decompensation, foveal burn, and corneal haze, as reported in the literature [11,56,86]. Significant complications, such as severe uveitis, IOP spikes that are more than 15 mmHg, etc., are contraindications for SLT repetition [87]. In this section, prevailing complications are described and case reports of sporadic serious complications are listed.

Iritis is a relatively common and mild complication occurring 2–3 days after SLT [56]. Damji et al. reported significantly lower incidences of the anterior chamber reaction in SLT compared with ALT [88]. Ayala et al. compared post-laser inflammation in the anterior chamber in patients with POAG with pseudoexfoliation (reported to be equal) [89].

Post-laser IOP elevation has been reported, ranging from 0% to 28% [38]. Latina et al. defined the IOP spike as 5 mmHg or more while Koucheki et al. defined IOP elevation as 6 mmHg or more and reported an IOP spike to be closely connected to the pigmentation extent of TM [7,56]. Harasymowycz et al. reported an IOP spike in their observational study of heavily-pigmented TMs and suggested special cautiousness with pigmentary dispersion syndrome and a heavily-pigmented TM [90].

Koktekir et al. reported severe bilateral anterior uveitis with posterior synechia, corneal haze, and endothelial loss after unilateral SLT, which proposes an autoimmune systemic response to be involved in the mechanism of action [91]. Systemic response in SLT is also supported in the findings by McIlraith et al.; they reported an IOP reduction in the untreated eye by 8% [92].

In a prospective study of 64 patients, evaluating macular thickness as measured by optical coherence tomography, the researchers did not find any significant increase in macular thickness after SLT [85]. However, there is one report of SLT-induced central macular edema and one report of worsening preexisting CME after SLT [93]. Wechsler and Wechsler reported a case of central macular edema after SLT [94]; nonetheless, it was a patient with preexisting CME and it was likely recidivant CME after topical therapy cessation rather than SLT-induced CME.

There were two cases of hyphema reported in the literature. The first case reported unilateral hyphema after bilateral SLT, which resolved spontaneously [95] and the second reported hyphema in a 77-year-old patient on topical and systemic NSAIDs [96].

In one case, choroidal effusion with narrow angles, and the other with milder previously described complications, developed after SLT, but were successfully treated and resolved [97]. While corneal edema occurs in 0.8% of cases [98], serious corneal complications, such as corneal haze and corneal melting, were reported [99,100]. An inflammatory cascade induced by SLT might reactivate herpes simplex infection, particularly in those patients on concomitant topical prostaglandin analogues [86]. An increase in central corneal thickness should also be considered in post-procedure IOP measurement [101]. There was one case of unilateral keratitis of unknown etiology after consecutive bilateral SLT [102]. Knickelbein et al. reported four cases of post-SLT corneal edema with subsequent thinning and a hyperopic shift, of which, two patients required contact lenses [103]. Special caution should be considered in treating post-LASIK patients. Holz and Pirouzian reported a case with bilateral diffuse lamellar keratitis after consecutive bilateral SLT [104].

Fortunately, severe complications are uncommon; nonetheless, they can threaten one’s sight. Therefore, they should be recognized, treated promptly, and all measures should be taken to avoid them [105].

## 9. Other Considerations: Retreatment, Predictors of Success, Cost-Effectiveness

The definition of SLT retreatment is somewhat ambiguous, because of variable protocols of 180-degree and 360-degree TM treatments. A repeat 180-degree approach could be considered a subsequent SLT in yet untreated TM. In the following studies, the 360-degree approach was used in repeating the SLT, which might be in fact considered as retreatment. Moreover, it was demonstrated that overlapping laser spots in a 180-degree SLT are linked to lower efficacy as compared to 360-degree nonoverlapping SLT [106]. Multiple studies have shown that SLT can be effectively repeated after the initial effect wears off [49,107]. In the LiGHT trial, it was demonstrated that if SLT is repeated as needed, the Kaplan–Meier survival estimates are better than if patients were managed with a single SLT treatment [47]. Repeating SLT in treatment naïve patients would thus yield far better IOP control in the long run. This was, to some extent, confirmed in a comparable real-life study by Ang et al. [108], where 45.7% of patients who maintained IOP-reduction at 24 months were treated twice. Another glimpse into real-world practice can be provided by a survey study by Canadian ophthalmologists on laser trabeculoplasty. A total of 87.1% of the participants thought of SLT as a repeatable procedure, mostly one or two repetitions [109]. In a retrospective study by Ansari, in the first year, 11% needed re-treatment; this increased to 40% at 5, and 58% at 10 years. Higher baseline IOP was significantly associated with an increased rate of retreatment and shorter retreatment times [80]. It was shown that repeating SLT in a timeframe shorter than one year after the initial treatment yielded a better success rate than if performed later [107]. Furthermore, the duration of success seemed longer after repeated SLT (13.1 months in comparison to 6.9 months after primary SLT) as shown by Avery et al. [110]. The notion of the added effect of second SLT was confirmed by the post hoc analysis of the SLT treatment arm in the LiGHT trial, where adjusted absolute IOP reduction was greater after SLT was repeated [47].

SLT seems to be generally accepted as effective; however, some patients in the studies performed better than others. Two recent studies [111,112] determined that pretreatment IOP was the only predictor of success after primary SLT. Hirabayashi et al. [113] stated that baseline IOP of >18 mmHg was significantly associated with increased success and that the IOP-lowering effect was greatest at 2 months and 6 months of follow-up. The effect of higher baseline IOP on success was confirmed in the real-world retrospective studies [83,84]. Khawaja et al. found that factors, such as glaucoma type or grade, TM pigmentation, or the type of topical medication, did not seem to predict SLT success. On the other hand, Abe et al. found such factors were associated with a lower risk for failure (denser angle pigmentation, corticosteroid treatment following SLT, and earlier stage glaucoma). The total energy delivered seem to have no role. The post hoc analysis of LiGHT demonstrated only two significant correlations: absolute IOP-reduction is positively predicted by higher IOP at baseline and slightly negatively by female gender [47]. It seems that patient selection based on predictors of success is yet to be fully comprehended; however, it appears that at a higher baseline, IOP could be the most significant. Recently, a retrospective study was published, examining the possibility of predicting the SLT outcome based on responsiveness to treatment with ripasudil drops. Ripasudil is one of the Rho-kinase inhibitors, which has distinct intracellular effects in the areas of tissue remodeling, fibrosis, and healing. It has a different mode of action compared to traditional medication in a way that it causes TM and Schlemm’s canal changes, resulting in a higher uveoscleral outflow, lowering IOP [114]. It was shown that patients who respond well to treatment with ripasudil had significantly better SLT success ratios compared to patients who were unresponsive to ripasudil treatment [115].

Glaucoma poses a significant economic burden, specifically due to population ageing. Cost-effective care is a major public health concern. A recent study conducted in the USA reported the highest eye-related costs for patients with OAG and OHT and determined positive economic externalities from therapies that delayed disease progression [116]. SLT is known as an effective method of lowering IOP and, thus, significantly partakes in lowering the economic burden of OAG.

Dirani et al. studied the economic effects of POAG in Australia and concluded that the use of laser trabeculoplasty as primary-line treatment rather than a second-line treatment would lead to a significant decrease of healthcare system costs [117]. Lee and Hutnik projected a 6-year cost comparison of primary SLT in therapy of OAG in Canada and found SLT to be cost effective [118].

During the LiGHT trial, cost-effectiveness in the UK was analyzed. They used a lifetime model, where cost-effectiveness was calculated in regard to cost per quality adjusted life year (QALY) of the SLT-first group, compared with the medicine-first group. The economic evaluation based on this trial determined that there is a 97% probability that SLT is a treatment for OAG and OHT, which is cost-effective [119]. This furthermore underpins findings that SLT as a first-line therapy is more economical when compared to hypotensive medication as the initial glaucoma therapy.

## 10. Future Perspectives and Alternatives Considered

As shown here, laser trabeculoplasty is an evolving field; using different lasers for trabeculoplasty and groundbreaking SLT treatment modifications might yield improved outcomes in the future.

A study by Gandolfi et al. [120] supports the concept that a 360-degree low energy SLT (0.3 mJ, 50–60 spots) could be repeated every year independently of measured IOP. It was shown that such patients remained medical treatment-free for 6.2 years. Based on these data, the COAST trial was launched to look at low-energy SLT in terms of TM anatomy and subsequent responsivity to SLT (awaiting results) [121]. If this treatment schedule proves to deter the use of medication or incisional surgery in the long-run, this might lead to further significant modifications in the field of treatment with SLT.

Transscleral SLT was a new modality of glaucoma laser treatment, first studied in Israel [122]. In essence, it means applying energy at the limbus, delivering the energy directly on the surface of the eye and not via gonioscopy lens. A standard SLT laser with modified parameters is used here. This approach proved effective, which was further studied in a prospective trial [123]. Laser energy delivered to the surface of the eye proved as efficient as standard SLT delivered to the TM via a gonioscopy lens. This is currently further studied in a multicenter prospective study [124] with an acronym GLAUrious, which tests direct transscleral SLT, delivered ab externo in POAG. The results are yet to be published. Transscleral SLT could potentially be useful in angle-closure glaucoma, where TM is not readily visible; however, separate trials are needed for evaluation. Recently automated transscleral SLT was studied. An automated image-processing algorithm targets predetermined targets at the limbus, automatically using a video camera, delivering transscleral SLT in 7 ns pulses. It proved to be easily performed, safe, and effective with up to 27% IOP reduction at 6 months, with a significant reduction in IOP-lowering medication [123]. Low-energy SLT and transscleral SLT were also included in the meta-analysis by Zhou et al. [31], where they were proven to be equally effective in lowering IOP when compared to medications as other laser trabeculoplasty procedures.

Recently, two reviews on the micropulse diode laser trabeculoplasty were published [125,126]. The reviews were conducted in a similar manner to SLT; however, a subthreshold micropulse diode laser technique as used that split up a continuous laser beam into on-and-off pulses to enable in-between cooling, similar to the micropulse modality of retinal micropulse laser treatment. The micropulse laser trabeculoplasty had initially shown similar comparable results to SLT in POAG [127]. The precise treatment protocol and laser wavelength are yet to be determined (by future prospective trials). Albeit it might be a safer treatment modality in regard to post-procedure complications, such as IOP-spikes or inflammation since trabecular structural change is less likely to occur [128].

Pattern scanning laser trabeculoplasty is a modality where the PASCAL laser is used, where computer-guided short pulse durations are used in 100 μm spots, presumably decreasing the surrounding tissue damage. In a RCT, this modality was performed in one eye and tested against SLT in the fellow eye—no significant difference in IOP-lowering was found at 6 months [129,130].

Titanium sapphire laser trabeculoplasty was compared to standard SLT in a RCT [131]. In this technique, near-infrared energy is used, which is believed to penetrate deeper to Schlemm’s canal and the ciliary body. No statistically significant differences in IOP-control or the success rate were noted, as well as no differences in the safety profiles.

## 11. Conclusions

More real-world studies with controls should be conducted to elucidate if the hype of SLT is real. The actual effectiveness of SLT alone was not entirely comprehended up until the LiGHT trial, where IOP-lowering was clearly demonstrated as at least equivalent to medication. In such settings, where SLT is used early in naïve patients, with higher initial IOP, it seems to be significantly more effective than when used as a later therapeutic choice. The latter supports the move of SLT up the chain of therapy in glaucoma in the new 5th edition EGS guidelines. Previously, patients might have been selected for SLT later, usually in between maximal medical therapy and surgery, at the bottom of the therapy algorithm. This might have been perceived as one of the reasons why retrospective real-world data were not as unequivocal in favor of SLT effectiveness in the long-run.

Currently, according to the available literature reviewed here and the EGS guidelines, SLT can be offered to patients as an alternative, where an initial topical therapy switch is considered or as an adjunctive therapy to the existing topical monotherapy. Be that as it may, we see SLT as a validated evidence-based alternative to medications, given as a first-line treatment in OAG and OH. This option will likely gain popularity amongst ophthalmologists in the future when more real-world data become available.

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
