# Peer review of "A Review of Selective Laser Trabeculoplasty: “The Hype Is Real”"

_jcm, 2022, doi:10.3390/jcm11133879_

Round 1

Reviewer 1 Report

This is a well conducted exhaustive review of a hot topic. The authors did a great job to summarize an impressive range of data and literature. I include some minors comments below

INTRO

·       Line 30 “treatment is therefore focused on lowering the intraocular pressure (IOP), which is the only risk factor, linked to glaucoma progression that can be successfully influenced”  rephrase to make it clearer. I suggest “which is the only modifiable risk factor”

·       Line 35 “In 1998 the first successful protocol of selective laser trabeculoplasty (SLT) was established, a 532 nm Q-switched frequency” It would be better to put one sentence about ALT in the intro

·       Line 48 “supporting the thesis for SLT as first in the line of” SLT is not a thesis, suggest replacing with “supporting the roleof” or “supporting the case for”

·       Line 51 society recently listed SLT higher in the algorithm of glaucoma treatment [13]. Would replace “higher” with earlier

BASIC PRINCIPLES

·       Line 73 “resistance, and thus restores normal outflow facility” . SLT improves outflow facility, it does not restore a normal facility

MECHANISM OF ACTION

·       Line 97 “analysis comparing ALT to SLT revealed better IOP reduction effects in SLT”. SLT and ALT have overall similar efficacy in terms of IOP lowering. Please refer to this quote from Alon, Skaat. “Selective Laser Trabeculoplasty: A Clinical Review.” Journal of current glaucoma practice vol. 7,2 (2013): 58-65. doi:10.5005/jp-journals-10008-1139.  The Cochrane Database Systematic review of LTP42,43 concluded in 2007 that there was some evidence to show similar efficacy in IOP control for SLT and ALT at 6 months and 1 year of follow-up. Since then, multiple retrospective44 and prospective45,46 clinical trials have been published and found no significant difference in IOP lowering when comparing SLT to ALT44,47,48 even over 5 years of follow-up.49 However, in retreatment SLT appears to lower IOP more effectively than ALT.47

·       Line 133 “it is also preferable to MIGS as a first step treat-133 ment [43,44]” This statement should be toned down a notch. The light study did not compare SLT to MIGS, the second paper cited here is not a randomized trial

·       Line 135 “The second group are patients with POAG or OH with uncontrollable IOP and disease progression that are already receiving glaucoma treatment”

SURGICAL TECHNIQUE

·       Line 171-178 “The complete glaucoma evaluation with ocular and medical history should be acquired. This should include visual acuity, applanation tonometry, central corneal thickness, static visual field and optical coherence tomography of the optic nerve head and the macula. etc” The authors suggest that all the above should be done prior to perform SLT. There is no consensus on what is a glaucoma evaluation so it would be better to keep it vague. Is it actually necessary to do a pachymetry and a macular scan before SLT? Better start this paragraph with the laser technique itself and skip the pre-laser exam (except the importance of gonioscopy). A thorough glaucoma evaluation including gonioscopy to assess if the patient is an eligible candidate for SLT

·       Line 179 and 193: Please avoid using words like “typically” state that there is no consensus on what is the best prophylactic prophylactic treatment fro IOP spikes and cite Zhang L, Weizer JS, Musch DC. Perioperative medications for preventing temporarily increased intraocular pressure after laser trabeculoplasty. Cochrane Database of Systematic Reviews 2017, Issue 2. Art. No.: CD010746. DOI: 10.1002/14651858.CD010746.pub2.

·       Line 183 “A 400-micron spot size and a 3 ns pulse duration is selected” Most SLT machines do not allow for a modification of spot size and duration

OUTCOMES

·       Line 310 “We believe that real clinical data, collected during every-day clinical practice is of 310 outmost relevance when validating the results from trials and metanalyses.” This statement suggest that personal experience is more important that randomized trials. It is true that real world studies/data are essential but this statement is misleading and should be nuanced or removed

·       Line 334 “In the latter study mostly white patients from UK were treated. It is likely that SLT is 334 more efficacious in patients with more pigmented TM [89], trait more commonly found in 335 nonwhite patients, although this is still unclear [13]. Mostly patients with more pigmented 336 TM have been included in the West Indies Glaucoma Laser Study, where treatment suc-337 cess of SLT was 78% at 12 months [90]” The only prognostic factor for SLT efficacy is baseline IOP. Race and pigmentation of TM are NOT linked to efficacy. Please remove any race related info from the following paragraph too

COMPLICATIONS:

When talking about keratitis please include herpetic keratitis as a possible complication especially in patients with a history of herpes keratitis.

OTHER CONSIDERATION

·       When talking about parameters and efficacy. Please include that overlapping spots are linked to lower efficacy (George MK, Emerson JW, Cheema SA, McGlynn R, Ford BA, Martone JF, Shields MB, Wand M. Evaluation of a modified protocol for selective laser trabeculoplasty. J Glaucoma. 2008 Apr-May;17(3):197-202. doi: 10.1097/IJG.0b013e3181567890. PMID: 18414105.)

CONCLUSION

·       Line 518: “More real-world studies with controls should be conducted to elucidate if the hype 518 of SLT indeed is real.”  The efficacy of SLT is already well established even as first line treatment. It is not a hype anymore but an essential part of the glaucoma algorithm. Please remove the “hype” concept from the rest of the paper. SLT was a hype twenty years ago, now it’s an evidence based treatment

·       Line 527 “Patients with higher baseline IOP could 527 have better response, however SLT might be equally successful in patients with other var-528 ious glaucoma-related characteristics. Nonetheless, patient selection still is crucial, thus 529 the actual use case of SLT is probably not justified across the board for OAG or OH patients, with considerations that efficacy is higher at early-stage glaucoma and higher initial 531 IOP, where up to 30% reduction of IOP was observed after treatment. Such a results is 532 comparable to hypotensive medication and could warrant use of SLT as first-line therapy, 533 nonetheless also in regards to its cost-effectiveness.” This paragraph is confusing and does not add to the conclusion. Please remove

·       Line 537 “Be that as it may, SLT can very likely compete with medications and even perform better given as a  first-line treatment. We see that as a probable OAG and OH initial treatment scenario in  the near future.” The scenario is not in the future but in the recent past year and in the present. Please modify the sentence to something like “SLT is a serious and evidence-based alternative to glaucoma mediation as a first line therapy in OAG and OH. This option is likely to gain popularity amongst ophthlmologists in the future when more real- world data becomes available”

Author Response

Thank you for your comments. We are very grateful for your time and consideration. The comments are point by point answered in bolded blue font.

This is a well conducted exhaustive review of a hot topic. The authors did a great job to summarize an impressive range of data and literature. I include some minors comments below

INTRO

  • Line 30 “treatment is therefore focused on lowering the intraocular pressure (IOP), which is the only risk factor, linked to glaucoma progression that can be successfully influenced”  rephrase to make it clearer. I suggest “which is the only modifiable risk factor”

Corrected

  • Line 35 “In 1998 the first successful protocol of selective laser trabeculoplasty (SLT) was established, a 532 nm Q-switched frequency” It would be better to put one sentence about ALT in the intro

We focused on leading the reader to the focus of this review as quickly as possible. While SLT is the main topic, ALT was quite thoroughly described in »Basic principles«. We believe this way we could add to the readability of this review. Hopefully you would agree with us.

  • Line 48 “supporting the thesis for SLT as first in the line of” SLT is not a thesis, suggest replacing with “supporting the roleof” or “supporting the case for”

Corrected

  • Line 51 society recently listed SLT higher in the algorithm of glaucoma treatment [13]. Would replace “higher” with earlier

Corrected

BASIC PRINCIPLES

  • Line 73 “resistance, and thus restores normal outflow facility” . SLT improves outflow facility, it does not restore a normal facility

We have corrected Line 73 in chapter Basic principles according to your suggestion and changed »restores normal outflow facility« to »improves outflow facility«.

MECHANISM OF ACTION

  • Line 97 “analysis comparing ALT to SLT revealed better IOP reduction effects in SLT”. SLT and ALT have overall similar efficacy in terms of IOP lowering. Please refer to this quote from Alon, Skaat. “Selective Laser Trabeculoplasty: A Clinical Review.” Journal of current glaucoma practicevol. 7,2 (2013): 58-65. doi:10.5005/jp-journals-10008-1139.  The Cochrane Database Systematic review of LTP42,43 concluded in 2007 that there was some evidence to show similar efficacy in IOP control for SLT and ALT at 6 months and 1 year of follow-up. Since then, multiple retrospective44 and prospective45,46 clinical trials have been published and found no significant difference in IOP lowering when comparing SLT to ALT44,47,48 even over 5 years of follow-up.49 However, in retreatment SLT appears to lower IOP more effectively than ALT.47

We have corrected Line 97 in chapter Mechanism of action according to your suggestion and changed »Meta-analysis comparing ALT to SLT revealed better IOP reduction effects in SLT« to »Meta-analysis comparing ALT to SLT revealed similar efficacy in therapeutic IOP response. However SLT results in a greater reduction of number of glaucoma medications versus ALT. Moreover SLT appears to be more effective in IOP reduction in retreatment versus ALT«. We added reference to this article: Alon, Skaat. “Selective Laser Trabeculoplasty: A Clinical Review.” Journal of current glaucoma practice vol. 7,2 (2013): 58-65. doi:10.5005/jp-journals-10008-1139.

  • Line 133 “it is also preferable to MIGS as a first step treat-133 ment [43,44]” This statement should be toned down a notch. The light study did not compare SLT to MIGS, the second paper cited here is not a randomized trial

We have corrected Line 133 »it is also preferable to MIGS as a first step treatment« to »it also has a comparable IOP reduction and complication profile to MIGS with smaller anatomical changes to the angle and can therefore be recommended as an alternative or a first step treatment.«

  • Line 135 “The second group are patients with POAG or OH with uncontrollable IOP and disease progression that are already receiving glaucoma treatment”

There is no written comment regarding line 135

SURGICAL TECHNIQUE

  • Line 171-178 “The complete glaucoma evaluation with ocular and medical history should be acquired. This should include visual acuity, applanation tonometry, central corneal thickness, static visual field and optical coherence tomography of the optic nerve head and the macula. etc” The authors suggest that all the above should be done prior to perform SLT. There is no consensus on what is a glaucoma evaluation so it would be better to keep it vague. Is it actually necessary to do a pachymetry and a macular scan before SLT? Better start this paragraph with the laser technique itself and skip the pre-laser exam (except the importance of gonioscopy). A thorough glaucoma evaluation including gonioscopy to assess if the patient is an eligible candidate for SLT

We have changed »The complete glaucoma evaluation with ocular and medical history should be acquired. This should include visual acuity, applanation tonometry, central corneal thickness, static visual field and optical coherence tomography of the optic nerve head and the macula. etc« to »To assess if the patient is an eligible candidate for SLT a thorough glaucoma evaluation has to be made before treatment. Special importance has to be given to gonioscopy, where the visibility and pigmentation of the TM have to be evaluated«

  • Line 179 and 193: Please avoid using words like “typically” state that there is no consensus on what is the best prophylactic prophylactic treatment fro IOP spikes and cite Zhang L, Weizer JS, Musch DC. Perioperative medications for preventing temporarily increased intraocular pressure after laser trabeculoplasty. Cochrane Database of Systematic Reviews 2017, Issue 2. Art. No.: CD010746. DOI: 10.1002/14651858.CD010746.pub2.

There were comments on this paragraph from other reviewers, it was therefore changed to »Studies have mostly shown that perioperative topical medication lower the risk of an IOP spike but there is no consensus on what the best prophylactic treatment is .  Most practitioners recommend using a topical alpha-adrenergic agonist (apraclonidine or brimonidine) 15 min to 60 min before treatment, some practitioners also use miotic drops (1% to 4% Pilocarpine).«  The provided citation was added.

  • Line 183 “A 400-micron spot size and a 3 ns pulse duration is selected” Most SLT machines do not allow for a modification of spot size and duration

Line 183 was changed to »A 400-micron spot size and 3 ns pulse duration are standard for SLT.«

OUTCOMES

  • Line 310 “We believe that real clinical data, collected during every-day clinical practice is of 310 outmost relevance when validating the results from trials and metanalyses.” This statement suggest that personal experience is more important that randomized trials. It is true that real world studies/data are essential but this statement is misleading and should be nuanced or removed

The statement was changed to: We believe that real clinical data, collected during every-day clinical adds to relevance of SLT to some extent when validating the results from trials and metanalyses.

  • Line 334 “In the latter study mostly white patients from UK were treated. It is likely that SLT is 334 more efficacious in patients with more pigmented TM [89], trait more commonly found in 335 nonwhite patients, although this is still unclear [13]. Mostly patients with more pigmented 336 TM have been included in the West Indies Glaucoma Laser Study, where treatment suc-337 cess of SLT was 78% at 12 months [90]” The only prognostic factor for SLT efficacy is baseline IOP. Race and pigmentation of TM are NOT linked to efficacy. Please remove any race related info from the following paragraph too

This paragraph and race-related data in the following paragraph were removed.

COMPLICATIONS:

When talking about keratitis please include herpetic keratitis as a possible complication especially in patients with a history of herpes keratitis.

We have included herpetic keratitis as a possible complication and added this sentence: »Inflammatory cascade induced by SLT might reactivate herpes simplex infection, particularly in those patients on concomitant topical prostaglandin analoges.«

OTHER CONSIDERATION

  • When talking about parameters and efficacy. Please include that overlapping spots are linked to lower efficacy (George MK, Emerson JW, Cheema SA, McGlynn R, Ford BA, Martone JF, Shields MB, Wand M. Evaluation of a modified protocol for selective laser trabeculoplasty. J Glaucoma. 2008 Apr-May;17(3):197-202. doi: 10.1097/IJG.0b013e3181567890. PMID: 18414105.)

We have added a sentence »Moreover it was demonstrated that overlapping laser spots in a 180-degree SLT are linked to lower efficacy as compared to 360-degree nonoverlapping SLT« and added the citation from George et al.

CONCLUSION

  • Line 518: “More real-world studies with controls should be conducted to elucidate if the hype 518 of SLT indeed is real.”  The efficacy of SLT is already well established even as first line treatment. It is not a hype anymore but an essential part of the glaucoma algorithm. Please remove the “hype” concept from the rest of the paper. SLT was a hype twenty years ago, now it’s an evidence based treatment

In an attempt to differentiate from the rest of the reviews the term »hype« was used as a non-scientific noun. As we have used it in the title of the review, we felt its informal use in the conclusion could be allowed. We can of course remove it without a compromise to the paper. Overall it does not relate to the method itself, more to its popularity. Please reconsider if you could accept the term being used in such a way.

  • Line 527 “Patients with higher baseline IOP could 527 have better response, however SLT might be equally successful in patients with other var-528 ious glaucoma-related characteristics. Nonetheless, patient selection still is crucial, thus 529 the actual use case of SLT is probably not justified across the board for OAG or OH patients, with considerations that efficacy is higher at early-stage glaucoma and higher initial 531 IOP, where up to 30% reduction of IOP was observed after treatment. Such a results is 532 comparable to hypotensive medication and could warrant use of SLT as first-line therapy, 533 nonetheless also in regards to its cost-effectiveness.” This paragraph is confusing and does not add to the conclusion. Please remove

Agree. Deleted.

  • Line 537 “Be that as it may, SLT can very likely compete with medications and even perform better given as a  first-line treatment. We see that as a probable OAG and OH initial treatment scenario in  the near future.” The scenario is not in the future but in the recent past year and in the present. Please modify the sentence to something like “SLT is a serious and evidence-based alternative to glaucoma mediation as a first line therapy in OAG and OH. This option is likely to gain popularity amongst ophthlmologists in the future when more real- world data becomes available”

Very good remark, we have restructured that. The last paragraph:

Currently, according to the available literature reviewed here and the EGS guidelines, SLT can be offered to patients as an alternative, where initial topical therapy switch is considered or as an adjunctive therapy to the existing topical monotherapy. Be that as it may, we see SLT as a validated evidence-based alternative to medications given as a first-line treatment in OAG and OH. This option is likely to gain popularity amongst ophthalmologists in the future when more real-world data becomes available.

Reviewer 2 Report

Summary: This review paper covers laser trabeculoplasty, its evolution and evolving role in glaucoma and ocular hypertension management, complications, and future directions including low energy and transscleral SLT.

Major comments: Overall, this is a thorough review that adds to prior reviews. Appreciate the discussion of literature with a neutral and balanced tone.

Minor Comments:

Line 40 Introduction- Can remove nevertheless in last sentence of first paragraph

Line 45 - Third sentence of 2nd paragraph – suggest “…somewhat ambiguous, [delete “and”] however it has appeared to be vastly more important in clinical practice than ascribed in the guidelines.”

Line 55 - Last paragraph of introduction – “..will outline some of [add “the”] crucial clinical guidelines for SLT”

Line 74 – trough -> through

Line 80 – capitalize Park

Line 87 – Perrish à Parrish

Line 98  – trough -> through

Line 179 and 194– perhaps provide a range of when alpha agonist is given as some providers instill 15 mins before laser and post-laser IOP check sometimes is done 30 mins after laser

Line 296 – “has reported no difference [in IOP or quality of life?] in treatments”

Line 311 - outmost change to utmost?

Line 424 – shorten to shorter

Line 471 – perhaps could add to title something like “Future Perspectives and Alternatives Considered” since last couple paragraphs discuss alternatives that were tried, but may not be considered strongly for future use given no significant improvement over SLT

Would appreciate if authors or journal can review grammar - such as line 431 two of recent studies à two recent studies,  line 445 – add “the” in front of possibility, line 446 – add “the” in front of rho-kinase inhibitors, line 454 add “A” in front “recent study”

Author Response

Thank you for your comments. We are very grateful for your time and consideration. The comments are point by point answered in bolded blue font.

Summary: This review paper covers laser trabeculoplasty, its evolution and evolving role in glaucoma and ocular hypertension management, complications, and future directions including low energy and transscleral SLT.

Major comments: Overall, this is a thorough review that adds to prior reviews. Appreciate the discussion of literature with a neutral and balanced tone.

Minor Comments:

Line 40 Introduction- Can remove nevertheless in last sentence of first paragraph

Corrected.

Line 45 - Third sentence of 2nd paragraph – suggest “…somewhat ambiguous, [delete “and”] however it has appeared to be vastly more important in clinical practice than ascribed in the guidelines.”

Corrected.

Line 55 - Last paragraph of introduction – “..will outline some of [add “the”] crucial clinical guidelines for SLT”

Corrected.

Line 74 – trough -> through

Corrected.

Line 80 – capitalize Park

Corrected.

Line 87 – Perrish à Parrish

Corrected.

Line 98  – trough -> through

Corrected.

Line 179 and 194– perhaps provide a range of when alpha agonist is given as some providers instill 15 mins before laser and post-laser IOP check sometimes is done 30 mins after laser

There were comments on this paragraph from other reviewers, it was therefore changed to »Studies have mostly shown that perioperative topical medication lower the risk of an IOP spike but there is no consensus on what the best prophylactic treatment is.  Most practitioners recommend using a topical alpha-adrenergic agonist (apraclonidine or brimonidine) 15 min to 60 min before treatment, some practitioners also use miotic drops (1% to 4% Pilocarpine).«  An appropriate citation was added.

Line 296 – “has reported no difference [in IOP or quality of life?] in treatments”

One of recent meta-analysis by Chi et al. [74] on SLT treatment in naïve patients versus medication with 1229 patients has reported no difference in treatments with SLT and medication-only treatments regarding the IOP reduction.

Line 311 - outmost change to utmost?

We have changed this sentence to: We believe that real clinical data, collected during every-day clinical adds to relevance of SLT to some extent when validating the results from trials and metanalyses. The comment however was valid, the spelling was incorrect, a lapsus linguae if you will.

Line 424 – shorten to shorter

Corrected.

Line 471 – perhaps could add to title something like “Future Perspectives and Alternatives Considered” since last couple paragraphs discuss alternatives that were tried, but may not be considered strongly for future use given no significant improvement over SLT

We have changed the title to »Future Perspectives and Alternatives Considered«, which we find very suitable. Thank you for this comment.

Would appreciate if authors or journal can review grammar - such as line 431 two of recent studies à two recent studies,  line 445 – add “the” in front of possibility, line 446 – add “the” in front of rho-kinase inhibitors, line 454 add “A” in front “recent study”

Corrected.